# Modeling the impact of COVID-19 on future tuberculosis burden

Mario Tovar [1,2✉], Alberto Aleta [3], Joaquín Sanz[1,2] & Yamir Moreno [1,2,3]

## Abstract

**Background** The ongoing COVID-19 pandemic has greatly disrupted our everyday life, forcing the adoption of non-pharmaceutical interventions in many countries and putting public health services and healthcare systems worldwide under stress. These circumstances are leading to unintended effects such as the increase in the burden of other diseases.

**Methods** Here, using a data-driven epidemiological model for tuberculosis (TB) spreading, we describe the expected rise in TB incidence and mortality if COVID-associated changes in TB notification are sustained and attributable entirely to disrupted diagnosis and treatment adherence.

**Results** Our calculations show that the reduction in diagnosis of new TB cases due to the COVID-19 pandemic could result in 228k (CI 187–276) excess deaths in India, 111k (CI 93–134) in Indonesia, 27k (CI 21–33) in Pakistan, and 12k (CI 9–18) in Kenya.

**Conclusions** We show that it is possible to reverse these excess deaths by increasing the pre-covid diagnosis capabilities from 15 to 50% for 2 to 4 years. This would prevent almost all TB-related excess mortality that could be caused by the COVID-19 pandemic if no additional preventative measures are introduced. Our work therefore provides guidelines for mitigating the impact of COVID-19 on tuberculosis epidemic in the years to come.

## Plain language summary

The COVID-19 pandemic has disrupted everyday life and put public health services and healthcare systems worldwide under stress. This has compromised the ability to control other diseases such as Malaria, Cancer and Tuberculosis. In this work we predict the rise in Tuberculosis occurrence and mortality when healthcare systems are impacted and diagnosis capabilities blocked in 4 countries where TB is prevalent. Our calculations show that an increase in new TB cases due to the COVID-19 pandemic could result in almost 400,000 additional deaths from TB in India, Indonesia, Pakistan and Kenya. We also show that increased diagnosis capabilities after the pandemic could reduce the additional deaths from TB resulting from the COVID-19 pandemic impact.

[1] Institute for Biocomputation and Physics of Complex Systems (BIFI), University of Zaragoza, 50009 Zaragoza, Spain. [2] Department of Theoretical Physics, University of Zaragoza, 50009 Zaragoza, Spain. [3] ISI Foundation, Via Chisola 5, 10126 Torino, Italy. ✉email: mariotovarzgz@gmail.com

Tuberculosis (TB) is an infectious disease caused by the bacterium *Mycobacterium tuberculosis* (*M.tb.*) that usually affects the lungs. It is a preventable but complex disease with a high global burden that requires early detection and long treatments. Despite the global effort to eradicate TB and recent decreases in its burden due to the implementation of strategies aimed at optimizing diagnosis and treatment[1,2], it remains one of the greatest threats to public health worldwide, being the deadliest single-agent persistent infectious disease nowadays. According to the 2021 Global TB Report by the World Health Organization (WHO)[3], ten million people developed TB and nearly 1.5 million people died because of TB infection in 2020, and for the first time in a decade, there is an increase in TB-caused deaths. In the last decades, the WHO has deployed a series of global strategies that have since been the backbone of the global fight against TB. In 1995, the Directed Observed Treatment Strategy was introduced, which significantly strengthened the capacity of national programs to diagnose and treat TB cases. Later, the Stop TB Strategy, announced in 2006, was the first of such plans at setting a TB elimination horizon, defined as a reduction of incidence levels under one case per million and year by 2050. A redefinition of the eradication goal took place in 2014 when the previous objective was moved forward to 2035 within the End TB Strategy.

If the elimination target set by the End TB strategy was already an ambitious goal[4], the emergence of the COVID-19 pandemic caused by the new coronavirus SARS-CoV-2 sheds significant concerns on whether these goals are still reachable. During the acute stages of the COVID-19 pandemic, economic and human resources were redirected to control and mitigate the emergency caused by the pandemic, which led to a great reduction in the diagnosis of new cases of other diseases, as already documented for cancer, or malaria[5,6]. Interventions such as long lockdowns and mobility restrictions have exacerbated shortages in resources otherwise destined for the care of patients suffering these, and other pathologies. Moreover, COVID-19 has greatly affected healthcare workers[7–9], thus creating additional pressure to healthcare systems.

TB diagnosis and patient care are no exceptions, as reported in previous literature[3,10,11]. As a primary and immediate effect of COVID-19 spreading onto TB transmission dynamics, a reduction in the case notification ratio has been observed during and immediately after lockdowns and periods of high COVID-19 incidence and saturation of healthcare facilities[3]. We hypothesize that this disruption alone will lead to a surge in TB burden in the next years, even before the more complex, and less predictable effects of the COVID-19 pandemic on TB management and transmission dynamics can be properly characterized. For example, drastic drops in laboratory capacity needed to support TB diagnosis are expected along with interruptions in the supply of drugs, which could result in shortages of medications and could delay the start of treatments until the supply chain is reestablished[12–14]. Moreover, as suggested by Cilloni et al.[15], even temporary stoppages might cause long-term increases in TB incidence and mortality, and a peak in TB burden is to be observed as a consequence of the healthcare system disruption.

In this work, we assess the impact of COVID-19 on the expected TB burden until the year 2035, which marks the target horizon of the End TB Strategy. Specifically, we incorporate the observed drop in TB diagnosis and treatment compliance rates caused by COVID-19 into a mathematical model that produces long-term forecasts of TB burden[16]. This allows us to: (1) quantify the effect of the COVID-19 stoppage in relation to a baseline scenario in which no pandemic happened, and (2) compute the effect that a rapid response to the uprising TB burden in the following years, in the form of a compensatory intervention aiming at boosting TB diagnosis rates as soon as the COVID-19 pandemic ends, has over long-term TB goals. Our results show that an effort focused on increasing TB diagnosis capabilities once the pandemic is over could revert the effect of the pandemic in the long term.

## Methods

**Model calibration and diagnosis rate.** In this study, we have capitalized on the detailed *M.tb.* transmission model developed by Arregui et al.[16,17] (see Supplementary Methods and Supplementary Fig. 1). Conceptually, this model is an age-stratified compartmental model that describes TB dynamics within a whole, closed population, stratified into 15 age groups during periods of the order of several decades. The model is detailed enough to include demographic evolution and aging, along with heterogeneous contact patterns among age groups that have been adapted from empirical survey studies (see Supplementary Fig. 2).

Here, the model is calibrated to reproduce TB incidence and mortality rates in each country under study for the period 2000–2019, using the burden estimates provided by the WHO. The calibration process gives the diagnosis rate $d(t)$ and the scaled infectiousness $\beta(t)$, which are modeled as half-sigmoid-like curves, and, among other parameters, are country-specific. This allows the model to reproduce different epidemiological scenarios. Specifically, the diagnosis rate is defined as:

$$d(t) = \begin{cases} d_0 + (d_{\sup} - d_0)t(t + \frac{1}{d_1})^{-1} & \text{if} \quad d_1 > 0 \\ d_0 & \text{if} \quad d_1 = 0 \\ d_0 - d_0 t(t - \frac{1}{d_1})^{-1} & \text{if} \quad d_1 < 0 \end{cases} \quad (1)$$

Therefore, the diagnosis rate is parameterized by two quantities $(d_0, d_1)$, where $d_0$ is the value at the beginning of the calibrating window (i.e., year 2000 in this study), and $d_1$ defines its evolution, either increasing or decreasing with time depending on $d_1$'s sign. In the case of a decreasing evolution, the diagnosis rate is bounded to be greater than zero, while in the case of increasing evolution the upper bound is $d_{\sup} = 12.17(\text{year}^{-1})$[16]. This latter upper bound corresponds to a minimum diagnosis period of 1 month, assuming that, with a conservative lower boundary, the main symptom of TB is a continuous cough lasting for 3 weeks, followed by a time to diagnose estimated to last at least 10 days[18]. For further details regarding the specific values of epidemiological parameters, calibration processes, and uncertainty estimates, the reader is referred to the original source[16] and the Supplementary Methods.

Once the model is calibrated, we use it to produce forecasts until 2035 under two different scenarios: the baseline scenario, namely, a scenario in which there is no COVID-19 pandemic and thus, no disruption in healthcare systems is introduced, and another one in which a disruption is introduced at the start of 2020 up to the end of 2021, which is the pandemic scenario. During the duration of the pandemic, the diagnosis rate drops according to the reduction observed in the notifications of TB cases that are reported by WHO online, in the last global TB report, and also by the Nikshay program in India. Therefore, the drops in diagnosis rate are country-specific. These drops in TB notifications are fitted to a bump-like asymmetric function, as described through $d_{\text{red}}(t)$ in Eq. (2). This function reproduces the real data and is then applied to the model-calibrated diagnosis rate to produce the diagnosis function under the pandemic scenario. The fitting procedure is a Levenberg-Marquardt Nonlinear Least-Squares using minpack.lm R's

**Table 1 Fitted parameters for diagnosis reduction in selected countries.**

**Bump parameters**

| Country | $\Theta = \{h, t_1, t_2, k_1, k_2\}$ |
|---|---|
| Indonesia | $\Theta = \{0.494, 0.391, 17.31, 6.226, 117.4\}$ |
| Pakistan | $\Theta = \{0.398, 0.257, 18.04, 3095.3, 826.8\}$ |
| Kenya | $\Theta = \{0.248, 0.859, 5.218, 0.905, 46.51\}$ |
| India | $\Theta_1 = \{0.393, 0.272, 1.25, 134.7, 0.645\}$ |
|  | $\Theta_2 = \{0.594, 0.139, 1.069, 3.111, 114.12\}$ |

The fitting procedure is a Levenberg-Marquardt Nonlinear Least-Squares using minpack.lm R's package[19], where Eq. (2) is applied to the WHO data normalized by the 2019 mean. For Indonesia, Pakistan and Kenya, one bump is enough for reproducing the data, whereas in India two separate bumps need to be concatenated, and are denoted here as $\Theta_1$ and $\Theta_2$ respectively. $h$, $k_1$ and $k_2$ are dimensionless quantities, whereas $t_1$ and $t_2$ have units of year[-1].

package[19], where Eq. (2) is applied to the data normalized by the 2019 mean for each country:

$$d_{\mathrm{red}}(t) = \begin{cases} x1 & t \le t_0 \\ 1 - h \cdot \exp\frac{-k_1(t-t_1)^2}{(t_1-t_0)^2-(t-t_1)^2} & t_0 < t \le t_1 \\ 1 - h \cdot \exp\frac{-k_2(t-t_1)^2}{(t_2-t_1)^2-(t-t_1)^2} & t_1 < t \le t_2 \\ 1 & t > t_2 \end{cases} \quad (2)$$

The bump-like function described in Eq. (2) serves as a multiplier to the model-calibrated diagnosis rate, thus, being the diagnosis rate under the pandemic scenario $D(t) = d(t) * d_{\mathrm{red}}(t)$ with $d_{\mathrm{red}}(t) \ne 1$ only during the COVID-19 pandemic, and $d_{\mathrm{red}}(t) = 1$ otherwise. In Table 1 we report the fitted values of each parameter involved in the bump-like description of the TB notification drops.

Finally, during the period of recovery, interventions are aimed at compensating for the drop in diagnosis rates during the pandemic years. We modeled this by multiplying the diagnosis by a scale parameter, as already discussed. Once the recovery period is over, we assume that the diagnosis rate goes back to its original value as given by $d(t)$ up to the end of the simulation.

**Modeling decisions about the disruption.** A conceptually deep limitation of this study that needs to be stressed is that we only describe the effects of COVID-19-induced reductions in TB diagnosis rates and treatment adherence as the main drivers of the interaction between both processes. Admittedly, the effects of the COVID-19 crisis on TB dynamics are more complex than what is described here, and will most likely include alterations in transmission dynamics, effects mediated by economic impact, and long-term damages to health care quality standards beyond diagnosis rates; all these being aspects that lie out the scope of our study, mainly because the relevant data needed to describe the effects of them on TB dynamics are yet to be produced.

Although some of the non-pharmaceutical interventions adopted worldwide have proven their efficacy in reducing COVID-19 spreading[20], their effectiveness highly depends upon general public adherence and proper knowledge about the pandemic risks. Whereas some studies[21–24] show that the knowledge, attitude, and practice toward COVID-19 basic preventive strategies and conducts are in general positive, there is a great variation between communities and, for example, in India, between socioeconomic levels. Specifically, rural populations, as well as individuals with lower education, and unskilled occupations, are associated with lower scores of knowledge, attitude, and practice toward the basic preventive strategies against COVID-19, which would in turn be expected to contribute to halting TB

transmission too[23]. This lack of adherence in the lower socio-economic levels[25] suggests that it might be misleading to assume that the implementation of countermeasures induces a reduction in the TB force of infection. On the other hand, the changes in mobility due to lockdowns and other restrictions indicate that most of the interactions happen in residential areas (e.g., households) while these interventions are in place (see Supplementary Notes 1 and Supplementary Fig. 3). Admittedly, this could be at the root of some recent observations that report that the number of children diagnosed with TB has increased and that non-pharmaceutical public health interventions likely reduced influenza transmission, but have a lesser effect on *M.tb.* transmission during 2020[26–28].

To contextualize our findings in broader scenarios where changes in TB transmission—either toward enhanced or reduced spreading—are considered, we show the results of the basic burden outcomes, incidence, and mortality, in each country, for scenarios in which the transmission is either reduced or enhanced. We also considered an alternative scenario in which the treatment availability is higher than the one adopted in the main text, thus, exploring the effect of an overestimate of the disruption over the treatment (see Supplementary Notes 2 and Supplementary Figs. 4 and 5).

**First-line treatment reduction.** According to previous reports[3,15], first-line TB treatment completion has dropped effectively as a consequence of the COVID-19 pandemic, with interruptions in the supply of drugs that delay the start of the treatment in those cases in which the remaining medical capabilities have been enough to diagnose the disease. This inconvenience could not only worsen the expected treatment outcome for the patient but also drive secondary infections even in diagnosed patients if they are not able to quarantine until the treatment could be carried out. We modeled this situation in terms of the epidemiological model by including a fraction of under-treatment pulmonary TB individuals ($T_p$) in the expression of the force of infection ($\lambda(t)$). On the baseline scenario and without disruptions, those $T_p$ individuals are not able to contribute to $\lambda(t)$ as we assume that they are under control by the healthcare system, thus, being controlled and either under quarantine or, later on, medicated with TB drugs that greatly reduce their infectiousness. This means that, under normal circumstances, diagnosed individuals are expected not to be a risk for the rest of the population. However, when disruptions in the supply chain appear, a drop is observed in the first-line and second-line treatments completion[15], and then diagnosed individuals who are not able to either start the treatment or quarantine could become a risk. For this reason, we obtain an estimate of the fraction of $T_p$ individuals that contribute to $\lambda(t)$, $T_{\mathrm{inf}}$, from Cilloni et al.[15] as:

$$T_{\mathrm{inf}} = (1 - \eta)T_p \quad (3)$$

where $\eta = 0.788$. This value attempts to capture this kind of impact in countries like India and Kenya. It is based on expert opinion in the Stop TB Partnership and USAID about the side effects of the COVID-19 pandemic on TB treatment completion. We assume it to be a good proxy for the real value for the other countries included in this study.

**Reporting summary.** Further information on research design is available in the Nature Research Reporting Summary linked to this article.

## Results

**Forecasts of TB incidence and mortality under COVID-19 pressure.** To forecast the effect that disruptions in the diagnostic capabilities and the treatment completion have on TB incidence and mortality trends, we selected four different high-burden

countries, three in Asia (India, Indonesia, Pakistan) and one in Africa (Kenya). Then, we calibrated the mathematical model[16] using the current WHO estimates for TB incidence and mortality rates in those countries and produced forecasts in two separate scenarios. The baseline scenario assumes no disruption, whereas the perturbed scenario incorporates the effects of the pandemic on TB diagnosis and treatment adherence. In the different countries analyzed, the duration of the disruptions has been of variable intensity and length, and while some countries experienced an almost complete return to pre-pandemic levels by June 2021 (India, Pakistan), other countries were still registering lower case notification rates by the end of 2021 compared to values before the COVID-19 irruption.

While treatment adherence is assumed to be reduced a 22% from the pre-pandemic values, according to Cilloni et al.[15], disruption is introduced based upon available data. These were made publicly available by the WHO for Indonesia, Kenya, and Pakistan[29], and by the Nikshay governmental program for India[30] during the months—or trimesters, for Pakistan—that followed the irruption of COVID-19. To incorporate those data into our model, we use a piece-wise bump function $d_{red}(t)$ to model a transient continuous drop in the diagnosis rate trend $d(t)$ that was foreseen within our model upon its calibration on pre-pandemic data (see "Methods", Eq. (1)). Proceeding this way, the actual diagnosis rate in the COVID-19 scenario, $D(t)$, can be obtained as the product of the model-calibrated diagnosis rate and the fitted bump function capturing the disruption due to the COVID-19 pandemic, as shown in Eq. (4) and Fig. 1:

$$D(t) = d(t) * d_{red}(t). \qquad (4)$$

Figure 2 shows the estimated TB incidence per million inhabitants per year in the four countries considered, both in the baseline scenario and considering the negative impact of the COVID-19 pandemic. As observed, a transient surge of TB incidence starts in 2020, which is later foreseen to return to values close to the baseline trend. In the figure, the dotted line represents the baseline scenario, namely, what would have been the projected evolution of TB incidence without the disruptions of the pandemic. The size of the peak reflects the severity of the saturation of the healthcare system in each country which led to drops of different intensity in diagnosis. The results show that the estimated COVID-19 impact on TB incidence trends is larger in the three Asian countries analyzed than in Kenya. This is a direct consequence of the less severe decays in TB case notifications that have been observed in Africa in comparison to other regions[3], which have been used to inform our mathematical model. These regional differences, in turn, may be due to a combination of factors. First, as stated by Haider et al.[31], some of those countries adopted early on measures for facing the pandemic, secondly, COVID-19 has had a smaller effect in Africa, which can be due, in part, to a strong under-diagnosis and partially because its younger population.

Important enough, even if COVID-19 disruptions are assumed to happen only during the pandemic years, the long-term effects span for longer times, sometimes up to 5 years since the start of the COVID-19 pandemic. As observed in Fig. 2, in the long term, TB incidence levels stabilize and recover to their baseline values approximately by the year 2030, resulting in a 10 years window of higher burden that makes the incidence go off the way of TB eradication stated in the End TB Strategy. Moreover, in the absence of any further intervention, the peak of TB incidence caused by the disruptions associated with the COVID-19 pandemic will produce not only new TB cases but also an increase in TB-related deaths all across the world. Specifically, by the end of the simulation period in the year 2035, our model predicts an increase in mortality as shown in Fig. 3, where we have represented both the increment percentage and the total number of accumulated additional deaths between 2020 and 2035. Particularly, we forecast an increase in the number of deaths of 1.28%(1.02–1.62, 95% CI) in India, 2.94%(2.50–3.54, 95% CI) in Indonesia, 0.72%(0.65–0.83, 95% CI) in Kenya and

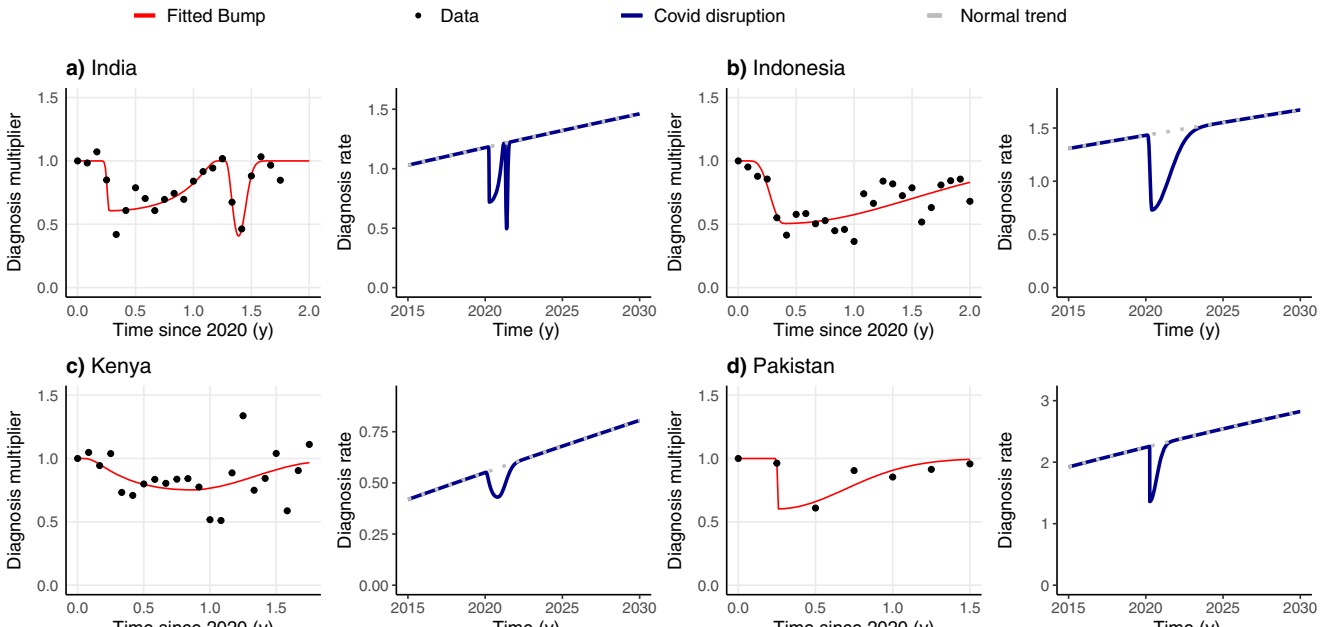

**Fig. 1 Changes in the diagnosis rate before, during, and after the pandemic period.** We present both the original data and the fitted bumps we use for modeling the disruption, along with the diagnosis rates in two scenarios: the baseline, with no disruption (dotted lines), and the pandemic scenario, where the drop in diagnosis happens and is followed by a return of the diagnosis rates to the baseline scenario. Diagnosis multipliers are obtained directly using the WHO data as the TB notifications in that period divided by the mean of TB notifications in the year 2019. The four countries considered are (**a**) India, (**b**) Indonesia, (**c**) Kenya, (**d**) Pakistan.

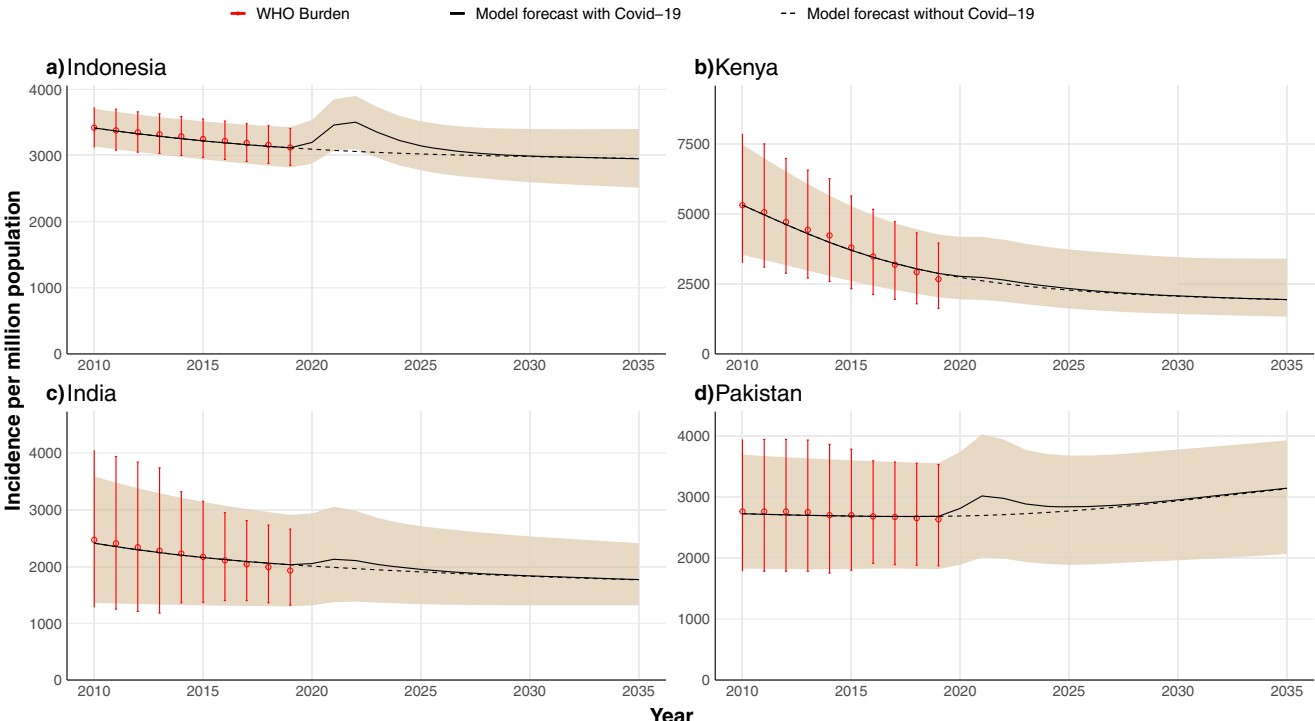

**Fig. 2 Projected annual TB incidence in four high-burden countries over the period 2020–2035.** The data-driven model is calibrated with WHO incidence data up to 2019[3]. The shaded area represents the 95% CI and the black line is the median of the model outcome for 500 independent runs of the disrupted scenario. The dotted black line is the model forecast for the scenario in which there was no Covid-19 pandemic. Red dots with error bars are the TB burden provided by the WHO[3] used for calibration. Projected incidence values are calculated at the end of the corresponding year on the *x*-axis. The impact of COVID-19 is modeled as a reduction in diagnosis rates and treatment completion for 2 years (2020 and 2021), see Fig. 1 and main text. The four countries considered are (**a**) Indonesia, (**b**) Kenya, (**c**) India, (**d**) Pakistan, which account for 42.1% of the total number of TB infections worldwide.

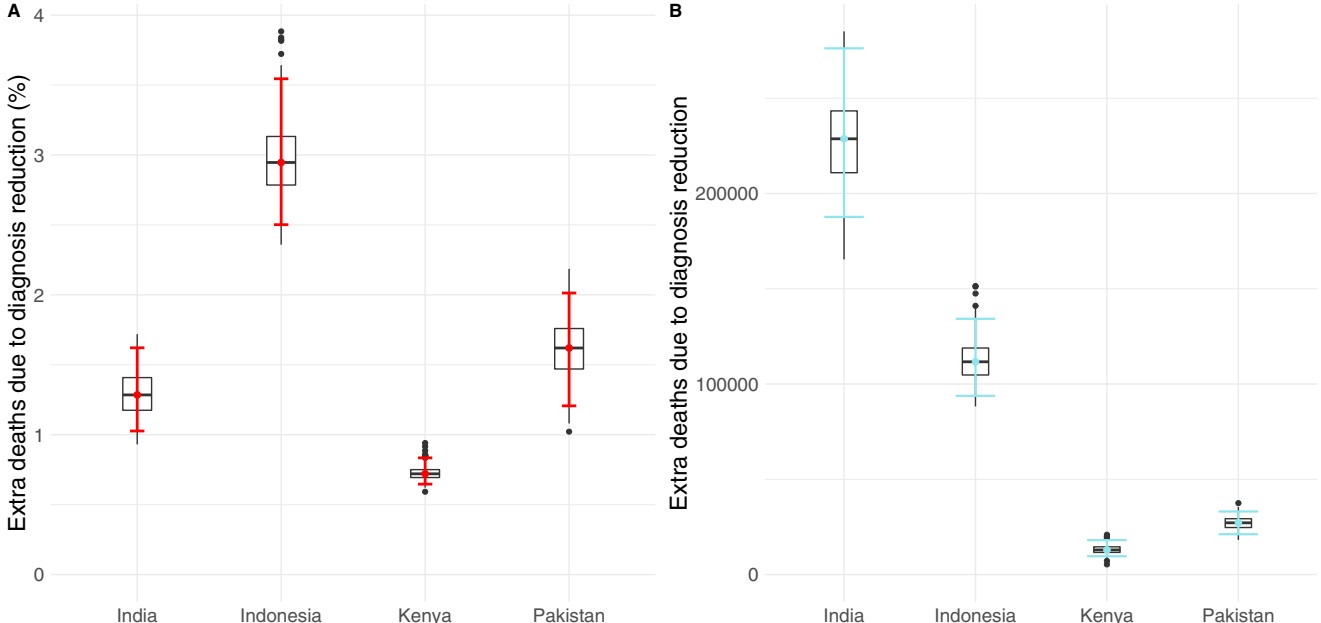

**Fig. 3 Model predictions of additional TB-related deaths due to COVID-19-related disruptions in healthcare systems.** Error bars are the 95% CI of 500 independent runs of the model. **A** Percentage of increase of mortality in comparison to the baseline scenario in 2035 for each of the four countries studied. **B** Cumulative number of excess deaths caused by the pandemic impact during the whole time window simulated (2020–2035) for each country under study as indicated in the *x*-axis.

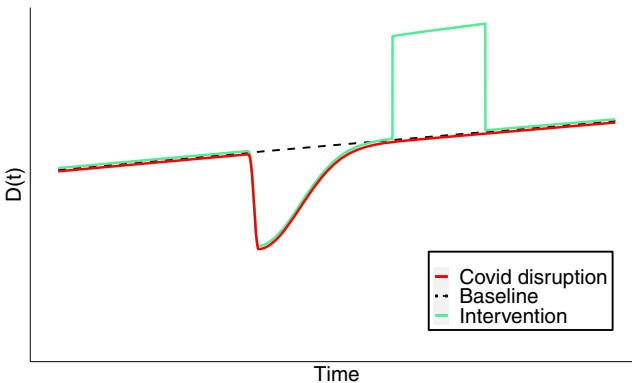

**Fig. 4 Schematic representation of the increase in diagnosis rate in the post-pandemic scenario.** Unlike the situation described in Fig. 1, we consider a compensatory period during which the diagnosis rate is boosted up to $d(t)*d_{inc}$, with $d_{inc} > 0$, which applies for the whole duration of the intervention.

1.62%(1.21–2.01, 95% CI) in Pakistan. In absolute terms, the total number of excess deaths could be over 350000 individuals in these four countries alone (Fig. 3B).

Finally, as the current situation with COVID-19 and its variants seems to be far from an end, we explore some alternative scenarios in which a secondary disruption, similar to the one that already happened, is introduced in the model. The results of this exploratory analysis are reported in the supplementary materials (see Supplementary Notes 3 and Supplementary Fig. 6).

**Proposed intervention for mitigating the pandemic effect**. As shown before, the pandemic circumstances will lead to notable increases in TB incidence and mortality. This represents a critical setback concerning the objective of eradicating TB disease within the next few decades, making it hardly achievable without a rapid and effective recovery strategy. More importantly, the disruptions will cause many preventable deaths. It is thus of utmost importance to elucidate whether new policies could be implemented to revert the negative impact of COVID-19 on TB disease. In what follows, we explore the potential of interventions focused on compensating the decay in diagnosis rates observed during the biennium 2020–2021, through a compensatory boost in the next years, as sketched in Fig. 4. While improvements in passive case finding routinely practice are unlikely to unlock sufficient increases in diagnosis rates, these, combined with the implementation of properly designed strategies of active case finding would constitute the paradigmatic type of interventions capable of producing diagnosis improvements comparable to those here explored.

The potential intervention over the diagnosis of TB cases is modeled using a piece-wise function that combines Eq. (4) with an additional piece introduced after the pandemic disruption is over, and for a parameterized duration, $T_{rec}^{end} - T_{rec}^{st}$, to be determined. More specifically, we assume that over this new period of time the pre-pandemic diagnosis rate is effectively increased by a factor $d_{inc} \geq 1$, (see also Fig. 4). That is:

$$D(t) = \begin{cases} d(t)*d_{red}(t) & \text{if} & t < T_{rec}^{st} \\ d(t)*d_{red}(t)*d_{inc} & \text{if} & T_{rec}^{st} \leq t < T_{rec}^{end} \\ d(t)*d_{red}(t) & \text{if} & t > T_{rec}^{end} \end{cases} \quad (5)$$

The efficacy of such a compensatory period will in principle be proportional to the intensity and the duration of the intervention. In Fig. 5, we show the impact of considering different combinations of diagnosis boost and duration of the boost on the cumulative excess mortality in 2035.

Clearly, the more intense and longer the additional effort is, the larger the number of averted deaths by the end of the simulation period. As it can be seen, an increase in diagnosis rate during a certain amount of time could eventually revert the negative impact of COVID-19 in TB mortality measured in 2035. More specifically, for all four countries, there is a region in the parameter space for which the increase in mortality in 2035 is close to zero (highlighted contour line of the different panels) if the diagnosis rate of new TB cases increases from 10 to 50% of its original value for a period that spans between 1 and 4 years. Importantly, this implies that the extra death toll that is expected from the effect of COVID-19 on TB diagnosis and treatment during the last 2 years could be fully mitigated if ambitious interventions focused on increasing case detection in the next few years are deployed.

Table 2 reports the number of averted deaths in each country in 2035 when the additional effort is applied for 2–4 years and considering increases in the diagnosis rates of 15, 30, and 45%, starting right after the end of 2022. The reported values are obtained by comparing model forecasts for the estimated number of TB-related deaths in the pandemic scenario with the outcome obtained when the recovery strategy is adopted after the end of the COVID-19 disruptions. The model suggests that it is generally better to increase the diagnosis rate for shorter times than to increase the temporal span and have smaller increments of the diagnosis rate. This is because, in the former situation, more deaths are averted in the long term. Nonetheless, the ideal scenario is still the one in which both dimensions are boosted at the same time, as the longer the time the effort is maintained for a given multiplier, the lower the TB-related death toll caused by the pandemic. As noted before, we stress that for an effort ratio of 1.30 and a temporal span of 4 years, the number of averted deaths almost corresponds to the 100% of additional deaths expected due to the COVID-19 disruptions (see Fig. 3), i.e., the pandemic impact on TB burden could be fully mitigated.

## Discussion

The COVID-19 pandemic is not over yet, and much remains to be clarified about its impact on the—physical and mental—health of the general population. As of May 2022, the coronavirus SARS-CoV-2 has infected more than 530 million individuals, causing the death of more than 6.3 million people worldwide. Although the SARS-CoV-2 and its associated disease COVID-19 were first identified more than 2 years ago, the scientific community has already been able to describe many of the clinical characteristics and pathogenesis of COVID-19, especially during the acute phase[32,33]. However, there are important features that remain less known, such as the long-term consequences of the disease[34–36] and the relation between comorbidities and their risks upon infection by SARS-CoV-2[37,38]. Another important question that is not fully assessed concerns the indirect effects of the pandemic, and the NPIs adopted for its control and mitigation, over other diseases.

In particular, the large number of healthy individuals that were infected in a very short period, producing the so-called epidemic waves, led to the saturation of many healthcare systems, which in turn induced the implementation of very restrictive measures such as lockdowns and curfews in those countries. These compulsory interventions have been argued to be at the root of important reductions in diagnosis rates of other deadly diseases[5,6]. Yet, the long-term consequences are still to be determined. Here, we have focused on TB, since it is one disease for which disruptions in health care could be most dramatic[13,15] given that even without a pandemic scenario, more than 1.5 million lives are lost every year because of the disease.

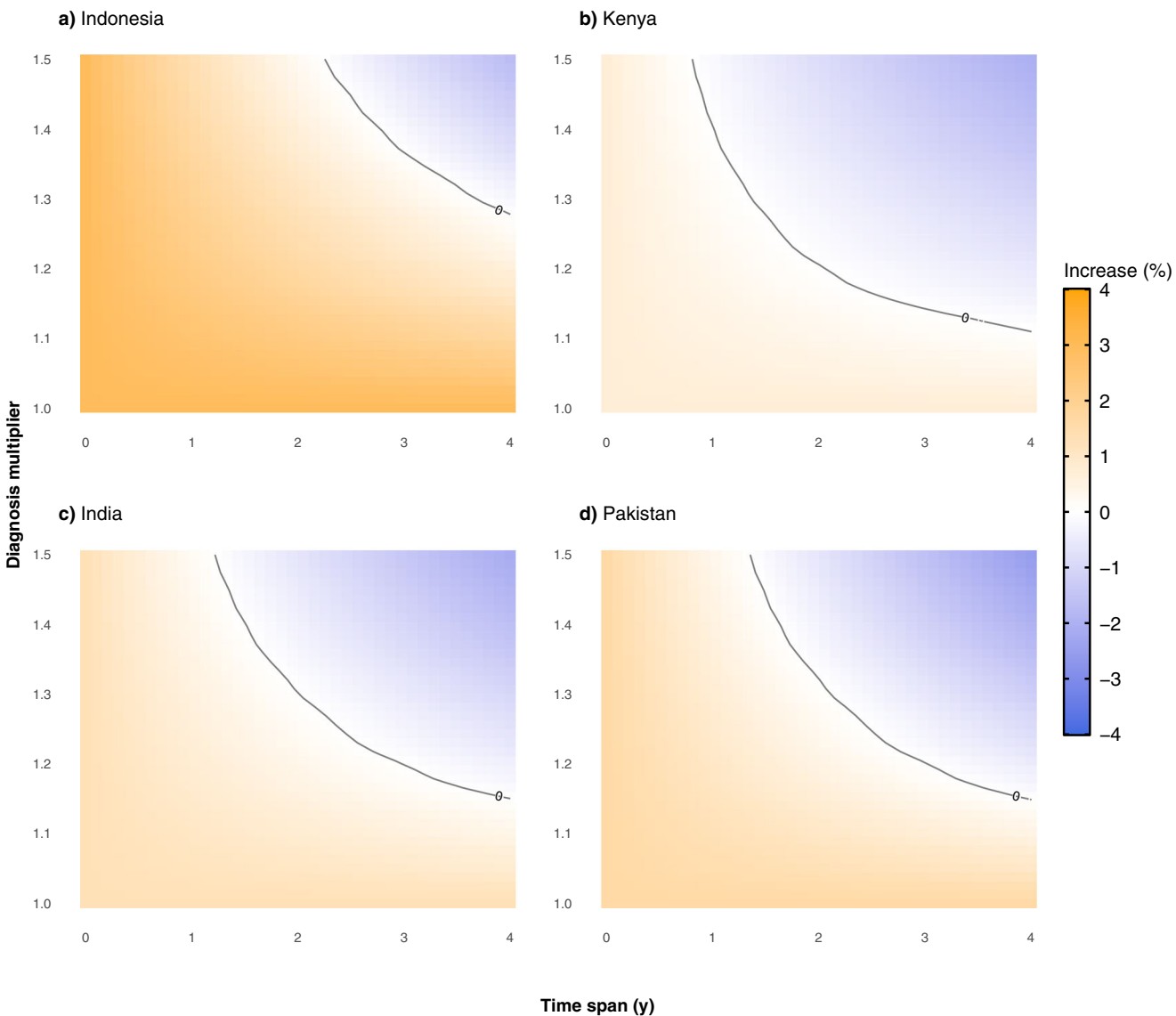

**Fig. 5 Relative increase of the expected number of deaths when an intervention is introduced in 2022 with respect to the projected impact of the COVID-19 pandemic.** We assume that during the recovery period, whose duration is given by the parameter $T_{rec}$ (the values of the *X*-axis), there is an increase in the diagnosis rate characterized by a factor $d_{inc} > 1$ (value of the *Y*-axis). Level (white) curves represent combinations of parameters that give the same excess deaths percentage, as indicated by the values over the curves. The four countries under study are (**a**) Indonesia, (**b**) Kenya, (**c**) India, (**d**) Pakistan.

Using a data-driven epidemiological model[16], we have quantified the negative impact of COVID-19 on TB diagnosis and its long-term consequences. We have also shown that a rapid and intense post-pandemic intervention could eventually mitigate the expected increase in incidence and mortality of TB. Countries enrolled in this work have been selected because of their high-TB burden, contributing an important amount of cases to the annual TB incidence recorded by the WHO global report. Certainly, all four countries together accounted for a 42.1% of the global TB cases in the year 2019. Individually, India comprises 26.5%, Indonesia accounts for 8.47%, Kenya represents 1.40% and Pakistan is responsible for 5.71% of all cases globally. Additionally, for these countries, the reduction in TB case notification due to the COVID-19 pandemic has been well-documented[3,29], and spans from milder (Kenya) to more severe magnitudes (Indonesia), which make them suitable case studies to estimate the pandemic negative impact and the design of corrective interventions.

Our results show that a drop in diagnosis rates and first-line treatment compliance statistics leads to a pronounced increase in TB incidence in comparison to the baseline scenario. In turn, the growth in TB burden leads to an upsurge in mortality, producing almost 400,000 excess deaths by 2035 in the four countries of the study combined. However, our study also shows that most of these deaths can still be prevented. In particular, our projections show that an increase of available diagnosis capabilities for some time has very positive effects on long-term TB burden since the pandemic effect can be greatly curtailed. More especifically, if the intervention is powerful and maintained for enough time, the entirety of the expected excess deaths can be avoided. It is worth stressing that the intervention proposed here is aimed at increasing the rate of diagnosed individuals, thus bringing them to treatment as soon as possible. This ultimately points toward cutting TB transmission to a point wherein pre-pandemic burden levels are recovered. As we have demonstrated here, one such intervention could be enough for full mitigation of the negative impact of the COVID-19 pandemic on TB incidence and mortality.

However, it is important to acknowledge that the specific interventions needed to achieve enhancements in the diagnosis

**Table 2 Cumulative number of averted deaths (in thousands) in 2035 with a post-COVID-19 intervention initiated in 2022 in each of the countries studied.**

**Number of averted deaths (in thousands) by 2035**

| Country | Diagnosis effort | T = 2 years | | T = 3 years | | T = 4 years | |
|---|---|---|---|---|---|---|---|
| Indonesia | 1.15 | 36 | (31–43) | 52 | (44–61) | 66 | (57–78) |
| | 1.30 | 67 | (57–79) | 93 | (80–110) | 118 | (102–139) |
| | 1.45 | 93 | (79–109) | 128 | (110–150) | 160 | (139–187) |
| Kenya | 1.15 | 9 | (7–13) | 13 | (10–19) | 17 | (12–24) |
| | 1.30 | 18 | (13–25) | 25 | (18–35) | 32 | (23–45) |
| | 1.45 | 25 | (18–35) | 35 | (26–50) | 45 | (32–63) |
| India | 1.15 | 121 | (98–156) | 173 | (141–224) | 223 | (182–288) |
| | 1.30 | 223 | (178–294) | 315 | (254–418) | 402 | (326–532) |
| | 1.45 | 309 | (246–419) | 432 | (347–586) | 547 | (444–735) |
| Pakistan | 1.15 | 14 | (12–16) | 20 | (17–23) | 27 | (23–30) |
| | 1.30 | 25 | (22–29) | 36 | (31–41) | 47 | (41–53) |
| | 1.45 | 35 | (30–39) | 49 | (43–56) | 64 | (56–71) |

The values in the table are computed by calculating the difference between the model forecast for mortality with the pandemic scenario and with non-pharmaceutical interventions of different intensities of diagnosis effort and duration of the recovery period. Values are the median of the outcome and figures in parentheses are the 95% CI of the model projections.

rates such as the ones explored in this study—between 10 and 50% increase above basal values—should most likely go beyond policies focused on reducing the diagnostic delay of patients seeking care after experiencing TB symptoms under passive case finding scenarios. Instead, active case finding strategies (ACF) constitute a robust family of interventions that can lead to reductions of both patient[39,40], and health care system components[41,42] of total TB diagnosis delays that are often compatible with case detection rate improvements similar to those explored here. Along these lines, several epidemiological studies published in the last decade report positive ACF experiences in diverse high-TB burden settings in both rural and urban areas in Africa and Asia alike. For example, in 2020, Vo et al. reported an increase of 15.9% of TB notifications (all-forms) in six districts of Ho-Chi Minh, Vietnam, with respect to another six control districts in the same city[43]. Other studies conducted previously in broad administrative districts in northern Uganda in 2018[44], and in Cambodia in 2016[45] report results from even more successful ACF strategies, able to increase all-forms TB diagnosis rates up to 30.4% and 46%, respectively, in comparison to control districts. Similar examples also include ACF strategies targeting rural, and even nomad populations, in African countries such as Ethiopia (ref. [46], 98.4% increase in all-forms of TB notification rates in 2013) or Nigeria (ref. [47], 24.5% of increase for all TB notifications among nomadic populations in Adamawa state, 2015). Finally, other studies have made use of mathematical modeling to stress that boosting diagnosis rates through ACF is not only feasible but also cost-effective in the mid to long term[48,49]. These results, taken together, suggest that the implementation of ambitious nationwide strategies of ACF in countries such as the ones studied here could contribute significantly to reducing the TB burden to the extent of mitigating the detrimental effects that COVID-19 has had on the TB epidemics worldwide.

In closing, we also mention that our approach is not exempt from limitations that affect TB transmission models. For instance, the outcome of our model depends on a series of epidemiological parameters and initial burden estimates that are subject to strong sources of uncertainty, thus propagating this uncertainty to the results. This means that future improvements in measuring the input data are expected to impact the quantitative outcomes of our mathematical model, in the same way as it would affect any other model that leans on them. Moreover, in our work, we have only described the disruption caused by the COVID-19 pandemic on the TB care system via a reduction of diagnostic capabilities and treatment availability. Even if these are arguably the primary, and the first effects of the COVID-19 pandemic on TB transmission dynamics that have been characterized, there may be many other effects that are yet hard to parameterize, such as the effect on the transmission that non-pharmaceutical interventions had in the countries that carried them out.

On the one hand, it should be possible, in the near future, to produce more detailed estimates on the disruptions of the pandemic on the complete TB cascade of care, based on the corresponding empiric data disclosed at a greater level of detail than the inputs used in this study. In fact, at the moment of writing this study, there is great heterogeneity in the available empirical data about the effects in the TB cascade of care, which points toward the urge of improving data availability for properly understanding the vast effects of COVID-19 on TB care[50]. Moreover, it is clear that other models, which provide a fine-grain structure capable of reproducing the full TB cascade of care, will be needed for taking advantage of this kind of data, which is certainly unfeasible with the model used in this study. Furthermore, it is well known that the emerging pandemic has disrupted profoundly the age structure of social contacts in human populations worldwide through a combination of mobility restrictions, lockdowns, social distancing, and adaptive conducts driven by self-perceived risk, often associated with the stark variations in susceptibility to severe disease and death that have been extensively reported for COVID-19. All these effects combined have arguably re-wired age-dependent contact structures in a way that is not fully understood, and may not be completely transient. On the other hand, geopolitical and economic shifts driven by the pandemic will for sure exert differential effects on TB transmission dynamics between countries. While the TB modeling community should commit to characterizing these phenomena in depth and incorporate them into model forecasts, these are all questions that remain beyond the scope of this study. Be it as it may, the model projections reported here point toward a worrying scenario about the effects of the current pandemic on TB burden evolution in the near future, regardless of the detailed implementation of the disruptions. As more data on the possibly disparate effects of the COVID-19 pandemic on TB is reported[3,29], updated modeling scenarios can be considered. Similarly, while the duration of the pandemic has been selected to be the length of the fitted bumps (which is directly related to available data) for all countries under study, the longer this data is

reported, the better quantitative outcomes can be forecasted. Similarly, precise measures of how the pandemic affects the model parameters, such as updated mortality risks or transmission rates, would also increase the quality of the forecasts.

In summary, our work shows that implementing a strategy aimed at boosting TB diagnosis rates after the pandemic holds the promise of mitigating, if not fully reverting, the negative impact of the COVID-19 pandemic on TB excess incidence and mortality, even if that period of boosted diagnosis is transient. While the importance of early diagnosis to arrest TB transmission is well known in TB epidemiology[51,52], we describe here how pushing that aspect of global TB management strategies in the early post-covid time has the potential of reverting a large fraction of the negative impact caused by the pandemic on the global TB epidemics. Interventions such as chronic cough screenings among people seeking healthcare, or even active screening of TB cases among non-symptomatic individuals, along with protocols targeting specifically pre-clinical and/or smear-negative TB cases do all hold the potential of boosting early diagnosis rates in a way that may well be compatible with the scenarios modeled in this study[51–54]. To prevent the COVID-19 pandemic from destroying all the progress achieved during the last years in global TB control, it is time to prioritize such interventions.

## Data availability
The data concerning drops in TB notifications and treatment availability are publicly available at the original sources[15,29,30]. The data underlying Figs. 1–5 is present in Supplementary Data 1–5. The data underlying Supplementary Figs. 3–6 is present in Supplementary Data 6–8.

## Code availability
The code that supports the findings of this study is available from the Zenodo repository[17] and at the following address https://doi.org/10.5281/zenodo.6638450.

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

## Acknowledgements
M.T. acknowledges the support of the Government of Aragón through a Government of Aragón Ph.D. contract order IIU/796/2019. M.T., A.A. and Y.M. acknowledge partial support from the Government of Aragon, Spain and "ERDF A way of making Europe" through grant E36-20R (FENOL), and by Ministerio de Ciencia e Innovación, Agencia Española de Investigación (MCIN/AEI/10.13039/501100011033) Grant No. PID2020-115800GB-I00. A.A. and Y.M. also acknowledge support from Soremartec S.A. and Soremartec Italia, Ferrero Group. J.S. acknowledges support from grants PID2019-106859GA-I00 and RYC-2017-23560 funded by MCIN/AEI/10.13039/501100011033 and by "ESF Investing in your future". The funders had no role in the study design, data collection, and analysis, decision to publish, or preparation of the manuscript.

## Author contributions
M.T., A.A., J.S. and Y.M. designed research; M.T. performed research; M.T., A.A., and Y.M. analyzed data; M.T., A.A., J.S. and Y.M. discussed results; M.T. wrote the first version of the paper with input from A.A. and Y.M.; and J.S. and Y.M. produced the final version of the manuscript. All authors read and approved the final version of the manuscript.

## Competing interests
The authors declare no competing interests.

## Ethical approval
The Declaration of Helsinki refers to "research on identifiable human material and data". This study does not use any individual human data. The public, aggregated statistics used in the study cannot be de-identified. Therefore, no ethics approval was needed.
