## [Peer Review File · Communications Medicine]

Reviewers' comments:

Reviewer #1 (Remarks to the Author):

This work is very interesting, which can provide guidelines for mitigating the impact of COVID-19 on tuberculosis epidemic in the years to come. However, the current work needs to be improved by a major revision.

1. Although the reference (14) has been provided in the study, more details on the models need to be provided as Supplemental materials.
2. For formula (1), it needs to be checked carefully.
3. As known, the current reductions in the TB incidence and mortality are mainly attributed to the lack of detection. Once the COVID-19 is ended, this effect will disappear, the TB epidemic will be prevalent in the last pattern. Whereas this forecast was done based on the current trend under the COVID-19 pandemic. So, this long-term forecast may be under-estimation for the epidemic trend. In order to make an accurate forecast, the reductions in the TB incidence during the COVID-19 outbreak should be added, then a forecast is done based on the amended data. This may be more reasonable.

Reviewer #2 (Remarks to the Author):

In this modeling manuscript, the authors use data on the initial reduction in TB notifications during the first several months of the COVID-19 to estimate the impact on medium-term TB incidence and mortality, and the magnitude of improvements in TB diagnosis that would be required to counteract this effect. While this is an important topic, I think there are several limitations of the current analysis that limit its relevance and make it prone to misinterpretation.

Note to the authors: You may recognize some comments from a previous review. I disclosed my history to the editor and was asked to review again. This review accounts for all updates made since I previously reviewed the manuscript.

My main concerns are as follows:

1. I don't think the pandemic's effects on TB treatment are estimated accurately. Specifically:
 - a. The authors mostly use data from WHO's 2020 global report and assume that the reductions in TB notification observed during the first few months of the pandemic will last for two years, however more recent data – including the 2021 version of WHO's report – show that notifications in many countries (including Kenya, Pakistan, and India) had returned to near baseline by early 2021. Patients who delayed treatment during initial lockdowns but remained symptomatic are likely to seek care before the pandemic is completely over. And anecdotally, TB programs have made modifications to improve care access and retention despite the strain of COVID. Thus, this analysis overestimates COVID's negative effects on TB diagnosis over the period modeled.
 - b. COVID's effects on treatment completion and resulting infectiousness also seem to be overestimated due to a misapplication of a published estimate. The authors are estimating that treated patients remain 0.788 times as infectious as those who have not been treated, due to

reduced treatment completion. Their source is expert opinion as summarized in Cilloni et al. It seems to me that the authors are misinterpreting and inverting the estimates from Cilloni et al, who estimated that treatment completion drops from ~90% to ~70% -- i.e., that treatment noncompletion makes treated individuals 22% (not 78%) as infectious as the untreated.

2. The intervention that is modeled – a boost in “diagnosis rates” – is infeasible and likely to be misinterpreted, because these diagnosis rates apply to all active TB (not only those who have symptoms or are seeking care). Thus, the modeled intervention would require dramatically increasing diagnosis of people with asymptomatic TB who are not yet in the health care system. The model structure groups all pulmonary TB disease into a single set of smear-stratified compartments, without differentiating by symptom or care-seeking status. Yet the authors assume that the time in this compartment (i.e. from disease onset to treatment) could be reduced to as little as a month. They justify this based on the typical duration of the classic TB symptoms, suggesting that they are talking about reducing the rate of diagnosis of symptomatic disease. Yet, half of prevalent TB is asymptomatic and the average case has active TB for many months before developing symptoms. Thus, the proposed improvements in diagnosis rate (e.g. a sustained 50% or 75% increase in the notification rate for multiple years) would require finding people with TB long before they develop symptoms or seek health care – a goal that is not feasible in current practice and that is not acknowledged in this manuscript. If this is going to be explored as a hypothetical intervention, then its programmatic implications (diagnosing most people before they develop symptoms, if the duration of TB is to be reduced to one month) and practical infeasibility need to be made more clear.

3. A major assumption in this analysis is that observed reductions in recent TB notifications are entirely due to underdiagnosis rather than reduced transmission. In reality, interventions that have contained the airborne transmission of COVID are almost certain to also affect TB transmission. Therefore, I ask the authors to more clearly contextualize their results in light of the assumption they have made. The authors do discuss unknowns about how COVID may affect TB transmission (changing contact networks, etc) as a general limitation, and their methods acknowledge that they are only considering effects on diagnosis and treatment completion -- but the specific assumption in their interpretation of notification data needs to be more clearly expressed in the abstract (e.g. “We describe the expected rise in TB incidence and mortality if COVID-associated changes in TB notification are sustained and attributable entirely to disrupted diagnosis”) and when presenting results.

4. In general, I’m not sure this is a good choice of underlying model for this research question. While a key strength of the preexisting model is its detailed representation of demographics, this structure is less useful for the current analysis without demographically-stratified data on COVID disruptions. Meanwhile, the model lacks granularity in representing the TB care cascade (care-seeking status, effects of incomplete treatment, repeated diagnostic attempts) that is relevant for understanding the cascade-related effects of COVID.

Reviewer #3 (Remarks to the Author):

This paper presents a model-approach to evaluate the mid- and long-term consequences of the COVID-19 pandemic on tuberculosis, both in terms of incidence and death toll. The paper is well written, and the results are very interesting for the evaluation of the world's TB situation in the following years. It is also worth that it proposes and evaluates a specific intervention to partially counteract such effects.

I hereby propose different items that could be taken into account by authors, in an attempt to increase its quality.

INTRODUCTION

The authors could consider to include in the Introduction two of the consequences of the COVID-19 pandemic on TB that have recently been published, in addition to the ones related with the diagnosis that are already discussed:

- Increase in TB-caused deaths for a first time in a decade (included in last WHO TB report)
- Shortage of anti-TB drugs (https://www.uitb.cat/wp-content/uploads/2021/11/nota-prensa-XXV-jornadas-TB_Eng.pdf)

MODEL CALIBRATION AND DIAGNOSE RATE

- Line 261: Please, mention that the model from [14] is an "age-stratified compartmental model", as it is essential to follow the subsequent explanation.
- Table II: I usually start reading the Methods before the results, since I assume that I'll be better prepared to understanding the Results. Then, I do not understand the "d_red(t)" in Table II, as for India. Please, explain it in the Methods. I saw afterwards that it is explained in the Results section, but it should be mentioned here as well.
- Figure 1: If d_red is a "multiplier" (see Table II caption and line 286), I'd expect the red line in the shadow area to follow an increasing trend parallel to the baseline instead of being flat. This is not the case according to the equation 1, so, the text in the Methods (Table II and line 286) should be better explained.

RESULTS

- Equation 1: is 2022 included in the middle term or in the bottom one?
- Line 109: you should mention that the less COVID-19 incidence in Africa can be also due, in part, to a strong under-diagnosis. In fact, young population can be infected as well, so, it can contribute to the incidence as other age ranges. Nevertheless, it is true that less serious cases are expected.
- Figure 2: in the legend, use a line instead of a circle for "Model". It would be useful to add the straight line for the disrupted scenario and the dotted line for the baseline model (in the legend, as well).
- Figure 4: I think that the figure would gain completeness if the red line showing the pandemic underdiagnosis is also shown, in another coloured area.

LIMITATIONS

- In lines 241-243 the authors mention that "the duration of the pandemic has been selected to be 2 years for all countries under study, longer estimates of this parameter could lead to an increase in the quantitative outcomes reported here.". Actually, given the current situation with omicron spreading at an enormous velocity worldwide, it is not reliable that the diagnosis rate is recovered by 2022. You should maybe extend such discussion. Have you evaluated the change in your results with an extra pandemic year? Can you provide an order of magnitude of such situation in terms of years to recover the baseline, for instance?

TYPOS

Line 299: Covid > COVID (also in caption of Figure 2)

Line 310: *Mycobacterium tuberculosis* > use italics

Reviewer #1 (Remarks to the Author):

This work is very interesting, which can provide guidelines for mitigating the impact of COVID-19 on tuberculosis epidemic in the years to come. However, the current work needs to be improved by a major revision.

We would like to thank the reviewer for his/her positive assessment of our work and for the suggestions and comments provided, which have helped us to improve the MS. We hope that the reviewer now thinks that our works is worth publishing in Communications Medicine. In the following paragraphs we include our answer to the main concerns highlighted by the reviewer.

1. Although the reference (14) has been provided in the study, more detailes on the models neede to be provided as Supplemental materials.

We agree with the reviewer. To solve this concern, the new version of the manuscript now includes more information about the TB model with two Supplementary figures that represent the natural history of TB in the model and the ageing and demographic evolution of the population. Additionally, we have included a whole new section that describes the system of Ordinal Differential Equations (ODE's) that are numerically integrated to solve the dynamics of each age-group inside the model.

2. For formula (1), it needs to be check carefully.

We thank the reviewer for pointing this out. In the new version of the manuscript, Eq. 1 has changed, as we have implemented a new methodology to account for drops in TB notifications. Now, we use the real data reported by the WHO to fit asymmetric bump-like functions that allow to reliably represent the data. These functions, which are multipliers to the diagnosis rate, equal 1 during the whole simulation except for when there is a disruption caused by COVID-19. This means that Eq. 1 is now:

$$D(t) = d_{red}(t) * d(t),$$

where $d_{red}(t)$ is the fitted bump-like function that represents the real data of changes in TB notifications, and that fulfils:

$$d_{red}(t) = \begin{cases} \neq 1 & \text{during the pandemic} \\ = 1 & \text{otherwise} \end{cases}.$$

This approach better represents reality and eliminates problems with the original Eq 1 and the previous approach.

3. As known, the current reductions in the TB incidence and mortality are mainly attributed to the lack of detection. Once the COVID-19 is ended, this effect will disappear, the TB epidemic will be prevalent in the last pattern. Whereas this forecast was done based on the current trend under the COVID-19 pandemic. So, this long-term forecast may be under-estimation for the epidemic trend. In order to make an accurate forecast, the reductions in the TB incidence during the COVID-19

outbreak should be added, then a forecast is done based on the amended data. This may be more reasonable.

We apologize for not being clear enough in the previous version of the manuscript. The model is first calibrated with all the data available previous to the COVID-19 pandemic outbreak (years 2000-2019). The forecast produced by the model in TB burden is based upon the incidence and mortality that is estimated from the model, and does not make use of the underreport in diagnosis or the reduced burden that arises as a consequence of it. Thus, regarding the diagnosis, we are not underestimating the forecast, as the expected incidence is always used to calculate the next integration step of the ODEs. We do agree with the reviewer, as we discuss within the limitations of the model, that there might be other factors that we do not take explicitly into account and that could eventually increase the estimates impact of COVID-19 in TB burden.

In the new version of the manuscript we updated a bit the results and methods sections to explain this better.

Reviewer #2 (Remarks to the Author):

In this modeling manuscript, the authors use data on the initial reduction in TB notifications during the first several months of the COVID-19 to estimate the impact on medium-term TB incidence and mortality, and the magnitude of improvements in TB diagnosis that would be required to counteract this effect. While this is an important topic, I think there are several limitations of the current analysis that limit its relevance and make it prone to misinterpretation.

Note to the authors: You may recognize some comments from a previous review. I disclosed my history to the editor and was asked to review again. This review accounts for all updates made since I previously reviewed the manuscript.

My main concerns are as follows:

1. I don't think the pandemic's effects on TB treatment are estimated accurately. Specifically:

a. The authors mostly use data from WHO's 2020 global report and assume that the reductions in TB notification observed during the first few months of the pandemic will last for two years, however more recent data – including the 2021 version of WHO's report – show that notifications in many countries (including Kenya, Pakistan, and India) had returned to near baseline by early 2021. Patients who delayed treatment during initial lockdowns but remained symptomatic are likely to seek care before the pandemic is completely over. And anecdotally, TB programs have made modifications to improve care access and retention despite the strain of COVID. Thus, this analysis overestimates COVID's negative effects on TB diagnosis over the period modeled.

We certainly thank the reviewer for this very pertinent comment, for we think she/he is right, and her/his criticism around this issue has allowed us to compile what we think is a rather improved, and more accurate modelling exercise in the resubmitted version of the manuscript.

In the previous version of our work, we used the same reduction in TB notifications at the start of the pandemic for a 2 years period, which, under the light of more recent data, turned out to be a clear over-estimation of the disruption caused by the pandemic on TB diagnosis capabilities in the countries under analyses.

In order to correct this issue, we now make use of the monthly-reported data concerning reductions in TB notifications from the WHO-TB database (ref. [17], <https://www.who.int/teams/global-tuberculosis-programme/data>), and adopt a data-driven approach. Specifically, we now use this data to fit bump-like asymmetric functions that model the diagnosis drops observed in each of the countries under analysis, and recalculate the diagnosis rate during the pandemic as the product of the model-calibrated diagnosis rate and the bump function, as it is explained in the new Figure 1, and in the methods section. As a result, we now describe, as foreseen by the reviewer, a significantly lighter effect for the COVID-19 related disruptions on TB burden trends, mediated by a more accurate, data-driven estimate of the reduction of diagnosis capabilities in the four countries under analysis.

b. COVID's effects on treatment completion and resulting infectiousness also seem to be overestimated due to a misapplication of a published estimate. The authors are estimating that treated patients remain 0.788 times as infectious as those who have not been treated, due to reduced treatment completion. Their source is expert opinion as summarized in Cilloni et al. It seems to me that the authors are misinterpreting and inverting the estimates from Cilloni et al, who estimated that treatment completion drops from ~90% to ~70% -- i.e., that treatment noncompletion makes treated individuals 22% (not 78%) as infectious as the untreated.

We think this is the result of a bad description of this aspect of our model, for which we apologize. We indeed report in the methods section that a fraction given by the 78% of the under-treatment individuals contribute towards infection, which is clearly not correct. Instead, is a fraction given by:

$$T_{inf} = (1 - \eta)T$$

which contributes in the force of infection, thus, being effectively a 22% of the individuals. In the simulations, we have been using $(1 - \eta)$ properly, but we described it incorrectly in the methods. This is now corrected in the new version of the manuscript, and we thank the reviewer for pointing this.

2. The intervention that is modeled – a boost in “diagnosis rates” – is infeasible and likely to be misinterpreted, because these diagnosis rates apply to all active TB (not only those who have symptoms or are seeking care). Thus, the modeled intervention would require dramatically increasing diagnosis of people with asymptomatic TB who are not yet in the health care system. The model structure groups all pulmonary TB disease into a single set of smear-stratified compartments, without differentiating by symptom or care-seeking status. Yet the authors assume that the time in this compartment (i.e. from disease onset to treatment) could be reduced to as little as a month. They justify this based on the typical duration of the classic TB symptoms, suggesting that they are talking about reducing the rate of diagnosis of symptomatic disease. Yet, half of prevalent TB is asymptomatic and the average case has active TB for many months before developing symptoms. Thus, the proposed improvements in diagnosis rate (e.g. a sustained 50% or 75% increase in the notification rate for multiple years) would require finding people with TB long before they develop symptoms or seek health care – a goal that is not feasible in current practice and that is not acknowledged in this manuscript. If this is going to be explored as a hypothetical intervention, then its programmatic implications (diagnosing most people before they develop symptoms, if the duration of TB is to be reduced to one month) and practical infeasibility need to be made more clear.

We would like to thank the reviewer also for pointing this question, for we essentially agree with her/him and we do think that thanks to this comment our resubmitted manuscript is more relevant, and less prone to misinterpretation.

We concur with the reviewer in that boosting diagnosis capabilities to the levels explored in this study can hardly be achieved through improvements of passive case finding practice. We also have to admit that our way to introduce and justify the ranges of the intervention explored in our previous submission (with boosts as large as 100% more than baseline trends) was probably misleading, and likely would have been (mis)interpreted the way the reviewer did, namely, as if such boosts in diagnosis rates could be achieved by implementing interventions aiming at reducing diagnosis delays for people seeking care after experiencing TB symptoms within passive case finding routine practice

alone. Since this is most likely unfeasible, as the reviewer is certainly right at pointing out, we have now introduced two modifications in the resubmitted version of the paper.

First, we have reduced the range of the interventions explored, now limited to 50% above baseline, which is one half the intensity of the interventions hypothesized in our original submission. Admittedly, this is also motivated by the fact that, since our estimates of the COVID-19 disruption on TB are now smaller, the intensity of the post-COVID interventions needed to counter arrest its effects are also smaller. We would like to highlight that our exercise describes the tradeoff between the time span of the intervention and its intensity, foreseeing that interventions of smaller effects on diagnosis rates, if sustained during enough time, would also translate into the mitigation of the extra mortality burden caused by the COVID-19, even if they were less acute. For example, if interventions were to be implemented during a time-span of four years, the percentage of diagnosis boosting that would be required to fully mitigate the COVID extra death toll would be only 30, 10, 15 and 15% in Indonesia, Kenya, India, and Pakistan, respectively.

Secondly, we have tried to leave clear that such diagnosis boosts, at a nation-level, are still unlikely to be achievable if they are bound to depend exclusively on interventions integrated within passive case finding routinely practiced. However, we argue, active case finding strategies implemented in geographic areas with millions of inhabitants have demonstrated to boost case finding rates to levels that are comparable, and sometimes larger, than those explored in this paper. These strategies have been implemented in different countries (Ethiopia, Nigeria, Uganda, Cambodia, Vietnam) both in rural and urban areas, and the conclusion that can be drawn from their analyses is that properly implemented ACF strategies could contribute to boost diagnosis rates at levels that are comparable to what is explored here in a cost-effective fashion. Therefore, in the discussion section of the resubmitted manuscript we introduce a brief discussion of how and why ACF may constitute a valuable tool to contribute to counter-arrest the COVID-19 pandemic effects on TB mortality, by boosting case finding rates similarly to what we describe here during the next few years.

Of course, scaling up these experiences to the nation-level in the countries analyzed is everything but trivial, and would require a significant economic and logistic effort by public health authorities and infrastructures in the countries analyzed. These are the programmatic implications, and limitations of the relevance of our analyses which we now try to discuss and acknowledge in the re-submitted version of the paper. Thanks again for the very valuable and precise criticism raised here.

3. A major assumption in this analysis is that observed reductions in recent TB notifications are entirely due to underdiagnosis rather than reduced transmission. In reality, interventions that have contained the airborne transmission of COVID are almost certain to also affect TB transmission. Therefore, I ask the authors to more clearly contextualize their results in light of the assumption they have made. The authors do discuss unknowns about how COVID may affect TB transmission (changing contact networks, etc) as a general limitation, and their methods acknowledge that they are only considering effects on diagnosis and treatment completion -- but the specific assumption in their interpretation of notification data needs to be more clearly expressed in the abstract (e.g. "We describe the expected rise in TB incidence and mortality if COVID-associated changes in TB notification are sustained and attributable entirely to disrupted diagnosis") and when presenting results.

We concur with the reviewer in that the fact that our modeling exercise of the effects of COVID-19 on the TB epidemics restricts to its documented effects on the diagnosis rates and treatment compliance levels should be stated more clearly as a study limitation, starting from the abstract. We have thus adopted his/her suggestion, and included the suggested statement in the abstract, pointing out that we model not just the disruption on the diagnosis, but also, on the treatment completion rates (see reviewer point 1b, for example).

Be it as it may, as the reviewer points out and we agree, in general the interventions that have contained the airborne transmission of COVID are likely to have also affected TB transmission. The problem however, is that nowadays there is likely a lack of quality supporting data to quantify -or even to estimate the direction, if we are honest and conservative enough- of these effects. The shifts in mobility trends towards indoors spaces observed during the pandemic, the relative lower adherence of lower socio-economic strata -those most affected by TB- to countermeasures such as mask usage (see refs. see refs [46-49]) as well as other contradictory observations when comparing interventions effects on influenza vs. M.tb. transmission (refs. [51-53]), delineate an admittedly complex overview for this question, which motivated us to adopt an agnostic approach and leave out of our model any effect of COVID-19 interventions on M.tb. transmission, as a sensible null-model-like in the lack of a clear evidence of the nature, intensity, or even the direction of such possible effects. We already discussed this thoroughly in the text, as well as in the supplementary appendix. However, in response to the reviewer comment, we have incorporated two changes in the resubmitted manuscript.

First, in the lack of empirical data supporting introducing in our model either a reduction or an increase in M.tb. transmission, we performed a sensitivity analysis on the changes in incidence and mortality as a function of the introduction of variations in M.tb. transmission that ranges between -15% and 15%. This analysis is included in the supplementary materials and discussed in the methods section B.

Second, we now tried to state more transparently that excluding eventual effects on transmission levels is indeed a relevant limitation of our study, and that disentangling the independent contribution of different counter measurements (e.g. mask wearing, mobility shifts, travel bans, school closures, etc.) adopted during the last two years on other airborne diseases such as TB should be a priority in epidemic modeling for the years to come (lines 209-218).

4. In general, I'm not sure this is a good choice of underlying model for this research question. While a key strength of the preexisting model is its detailed representation of demographics, this structure is less useful for the current analysis without demographically-stratified data on COVID disruptions. Meanwhile, the model lacks granularity in representing the TB care cascade (care-seeking status, effects of incomplete treatment, repeated diagnostic attempts) that is relevant for understanding the cascade-related effects of COVID.

We agree with the reviewer that our model has important structural limitations that might prevent us from using it to explore further effects of COVID-19 on TB dynamics, based on eventual additional empiric data at a higher level of detail than what we consider here (e.g. effects of COVID-19 at the different stages of the TB cascade of care).

Indeed, we acknowledge that explicitly in the discussion section (lines 209-241), and this is precisely the reason why we have just avoided any kind of exercise in that direction, and limited ourselves to

provide a description of the effects of COVID-19 pandemics on TB detection/notification rates, and to treatment adherence, which we certainly can do with our model. Conducting such kind of “fine-grain” modeling exercise in relation to COVID-TB interactions remains a pending question -not only for the sake of our work, but for the entire community. This is in part because the data available about the multiple disruptions that have already been described (reviewed in ref. [38]), is highly heterogeneous in nature, and only available for specific settings. This perhaps had not been correctly framed in our previous version of the manuscript, which we now try to correct in lines 219-225 of the discussion.

However, we must disagree in the observation that our model constitutes an inadequate tool to present the modeling exercise we do here. While it has for sure its own limitations as we discuss in detail above, and in the manuscript, it has also advantageous features to complete the task we do here. As an example, modelling diagnosis rates as dynamical variables varying smoothly with time following a country-wise sigmoid that is fitted from a training period (years 2000-2019), with a time-step of less than one natural day constitutes a rather adequate trait to accommodate -now in the four countries under scrutiny -modelling the disruptions of COVID-19, using bump-functions as suitable continuous functions fitted from monthly data-, from a data-driven perspective.

Admittedly, without this trait, we would not have been able to provide a response to reviewer`s point #1, or at least not one based on actually incorporating the different quantitative, continuous trends for the disruptions observed in each country. This is certainly a feature that other models in recent TB literature lack, even models that have been used to conduct similar exercises to the one presented here, related to the COVID-19 disruptions on the TB epidemics (see for example [11], based on a model which, for example, has a minimum time-resolution of six months, and also do not provide much of a description of the cascade of care).

In short, we argue that, while it is totally true that our model lacks granularity, and, in general degree of detail needed to describe certain important aspects of the problem that lie beyond the scope of our study (such as the effects of COVID-19 on the different levels of the cascade of care), it does provide a description of TB dynamics whose level of detail *is adequate in* relation to the specific phenomena that we are modeling, that is, the effect of COVID-19 on case finding rates and treatment adherence.

To clarify this question, we have now introduced a paragraph in the discussion (lines 219-223) highlighting the need to integrate detailed information on the effects of COVID-19 on TB dynamics at the different levels of the cascade of care within mathematical modelling frameworks to fully understand the effects of the new virus on the old disease.

In general, we would like to thank the reviewer for the tremendous value of her/his comments, and the positive predisposition towards the -iterated- revision of our work.

Reviewer #3 (Remarks to the Author):

This paper presents a model-approach to evaluate the mid- and long-term consequences of the COVID-19 pandemic on tuberculosis, both in terms of incidence and death toll. The paper is well written, and the results are very interesting for the evaluation of the world's TB situation in the following years. It is also worth that it proposes and evaluates a specific intervention to partially counteract such effects.

I hereby propose different items that could be taken into account by authors, in an attempt to increase its quality.

We would like to thank the reviewer for his/her positive assessment of our work and for the suggestions and comments provided, which have helped us to improve the MS. We hope that the reviewer now thinks that our works is worth publishing in Communications Medicine. Please, find below our detailed responses.

INTRODUCTION

The authors could consider to include in the Introduction two of the consequences of the COVID-19 pandemic on TB that have recently been published, in addition to the ones related with the diagnosis that are already discussed:

- *Increase in TB-caused deaths for a first time in a decade (included in last WHO TB report)*
- *Shortage of anti-TB drugs (https://www.uitb.cat/wp-content/uploads/2021/11/nota-prensa-XXV-jornadas-TB_Eng.pdf)*

We thank the reviewer for sharing these two references that we missed. We agree that they are relevant and in the new version of the manuscript we included both of them.

MODEL CALIBRATION AND DIAGNOSE RATE

- Line 261: Please, mention that the model from [14] is an "age-stratified compartmental model", as it is essential to follow the subsequent explanation.

We agree, this is now added.

- Table II: I usually start reading the Methods before the results, since I assume that I'll be better prepared to understanding the Results. Then, I do not understand the " $d_{red}(t)$ " in Table II, as for India. Please, explain it in the Methods. I saw afterwards that it is explained in the Results section, but it should be mentioned here as well.

This table has changed in the new version of the manuscript, as we model COVID-19 disruption using bump functions that are fitted to the real data. Table II now contains the values of the calibrated parameters for the fitted bumps.

- Figure 1: If d_{red} is a "multiplier" (see Table II caption and line 286), I'd expect the red line in the shadow area to follow an increasing trend parallel to the baseline instead of being flat. This is not

the case according to the equation 1, so, the text in the Methods (Table II and line 286) should be better explained.

Similarly, as for the previous concern, this has changed as now we do not model COVID-19 as a flat disruption during the whole period. Instead, we use a fitted asymmetric bump function that models the real drop in notifications, and the methods section is now updated with this new methodology.

RESULTS

- Equation 1: is 2022 included in the middle term or in the bottom one?

In the new version of the manuscript there is no need for this, and Eq. 1 is updated to include the bump function, whose end depends upon data instead of being arbitrarily chosen.

- Line 109: you should mention that the less COVID-19 incidence in Africa can be also due, in part, to a strong under-diagnosis. In fact, young population can be infected as well, so, it can contribute to the incidence as other age ranges. Nevertheless, it is true that less serious cases are expected.

We agree and thank the reviewer for pointing this out. It has been added in the corresponding section.

- Figure 2: in the legend, use a line instead of a circle for "Model". It would be useful to add the straight line for the disrupted scenario and the dotted line for the baseline model (in the legend, as well).

This is right, Figure 2 is updated and we have corrected the legend.

- Figure 4: I think that the figure would gain completeness if the red line showing the pandemic underdiagnosis is also shown, in another coloured area.

We agree and in the new version we have updated Figure 4 to reflect not only the intervention boost but also the disruption modelled as a bump.

LIMITATIONS

- In lines 241-243 the authors mention that "the duration of the pandemic has been selected to be 2 years for all countries under study, longer estimates of this parameter could lead to an increase in the quantitative outcomes reported here.". Actually, given the current situation with omicron spreading at an enormous velocity worldwide, it is not reliable that the diagnosis rate is recovered by 2022. You should maybe extend such discussion. Have you evaluated the change in your results with an extra pandemic year? Can you provide an order of magnitude of such situation in terms of years to recover the baseline, for instance?

The reviewer is right in her/his observation that all our analyses, in the first submission, dealt exclusively with the case of COVID-19 lasting for two years, which regarding the actual situation, seemed too optimistic. To solve these concerns, we implemented the following changes in the modified version:

- We added a paragraph in the discussion about this problem.

- In the Supplementary materials, we provide a new sensitivity analysis in which we explore the evolution of TB burden with a new disruption in TB notifications, which we assume has the same functional form as the fitted bumps, and we recalculate the incidence and mortality in the new scenarios. (i.e., we repeated the same drop in notifications that countries suffered just at the end of the real bump, and in the case of India, both scenarios, one per bump, are analyzed, and the TB burden is then forecasted.)

TYPOS

Line 299: Covid > COVID (also in caption of Figure 2)

Line 310: Mycobacterium tuberculosis > use italics

Thanks. These are now corrected.

REVIEWERS' COMMENTS:

Reviewer #1 (Remarks to the Author):

the authors have put additional efforts to address the concerns raised, and therefore I agreed to accept this original manuscript.

Reviewer #2 (Remarks to the Author):

The authors have responded adequately and thoughtfully to my comments and made appropriate, substantial revisions to their manuscript. I have no remaining concerns.

Reviewer #3 (Remarks to the Author):

I acknowledge the efforts of the authors for addressing our comments, which I think have increased the manuscript's quality.

I'd like the authors to add the units of the parameters listed in Table II.